# Surface-Functionalized Silver Nanoparticles Boost Oxidative Stress and Prime Potatoes Against Phytopathogens

**DOI:** 10.3390/plants15020203

**Published:** 2026-01-09

**Authors:** Alexey A. Kudrinsky, Dmitry M. Mikhaylov, Olga A. Shapoval, Georgii V. Lisichkin, Yurii A. Krutyakov

**Affiliations:** 1Department of Chemistry, M. V. Lomonosov Moscow State University, Moscow 119991, Russia; lisich@petrol.chem.msu.ru; 2Software Engineering, Institute of Computer Technologies and AI, Kyrgyz National University Named After J. Balasagyn, Bishkek 720000, Kyrgyzstan; dmitry.mikhaylov@knu.kg; 3Impactive Technologies Holding Limited, Abu Dhabi P.O. Box 95044, United Arab Emirates; 4Department of Testing Growth Regulators and Agrochemicals, Pryanishnikov All-Russian Scientific Research Institute of Agrochemistry, Moscow 127550, Russia; shapowal.olga@yandex.ru

**Keywords:** induced resistance, catalase, peroxidase, priming, silver nanoparticles (AgNPs), potato

## Abstract

The study investigates the use of silver nanoparticles (AgNPs) in agriculture, focusing on their potential to enhance the immune response of potato (*Solanum tuberosum* L.) plants against phytopathogenic attacks. The research highlights how AgNPs, stabilized by biologically active polymers polyhexamethylene biguanide and tallow amphopolycarboxyglycinate, can induce oxidative stress. Triple foliar application of 0.1–9.0 g/ha silver nanoparticles at the budding and later stages demonstrated significant efficacy in suppressing diseases caused by *Phytophthora infestans* and *Alternaria solani* (over 60%). This effect was linked to the increased activity of peroxidase—over 30–50%—and the decreased catalase activity, indicative of a well-coordinated oxidative stress response to the invasion of *P. infestans* and *A. solani*. The results suggest that AgNPs in low concentrations can prime the plant’s innate immune system, enhancing its resistance without detrimental effects on growth parameters, thus contributing to the improved crop yield. These findings underscore the potential of AgNPs not as traditional biocides, but as intelligent elicitors of plant-induced resistance, positioning them as next-generation tools for sustainable crop protection and yield optimization, which can be applied at extremely low doses (less than 10 g/ha of active substance).

## 1. Introduction

Modern agriculture faces the dual challenge of ensuring food security while minimizing its environmental footprint. In order to achieve long-term intensification in agriculture, a paradigm shift from conventional pesticide use towards innovative strategies that bolster crop resilience is needed. In this regard, development of next-generation agents that do not directly target pathogens, but rather act as intelligent elicitors, priming the plant’s immune system—a concept central to induced resistance—shows a lot of promise. This approach can significantly reduce the active substance input per hectare, aligning with strict ecological and regulatory demands.

The cornerstone of plants’ innate immune response is the rapid and localized production of reactive oxygen species (ROS), a phenomenon known as the oxidative burst. This tightly regulated oxidative stress serves as a critical signaling hub, orchestrating downstream defenses such as the reinforcement of cell walls and the induction of localized programmed cell death and the hypersensitive response (HR), which confines biotrophic pathogens [1,2,3,4,5,6,7,8,9,10,11]. Thus, targeted modulation of the ROS network represents a strategic lever for enhancing phytoprotection.

Notably, oxidative stress can also be induced by certain agrochemicals. Traditional copper-based fungicides, for instance, are known to promote ROS generation via Fenton-like reactions. Similarly, engineered silver nanoparticles (AgNPs), which are increasingly studied in relation to their antimicrobial and plant growth-inducing properties, have been proven to interact with cellular components, potentially disrupting redox homeostasis and amplifying oxidative stress in plant tissues [12,13,14]. This shared pathway creates a complex interplay: both pathogen invasion and metal-based treatments converge on ROS-mediated signaling. Consequently, their combined effect on the plant’s defensive metabolism can be synergistic, antagonistic, or context-dependent, making empirical investigation essential.

Critically, prior research has largely been focused on the independent effects of either pathogen challenge or AgNP exposure, often employing AgNPs at concentrations where direct biocidal activity is observed. In contrast, the potential of AgNPs applied at extremely low, non-fungicidal doses—previously explored mainly in relation to growth stimulation—to act as resistance primers under genuine disease pressure remains underexplored. This gap highlights the need to study their role not as mere substitutes for pesticides, but as specific modulators of plant immunity in the phytopathological context.

Therefore, the present study was designed to highlight how simultaneous biotic stress and treatment with polymer-stabilized AgNPs, at doses devoid of direct fungicidal effect, influence the oxidative stress landscape in potato (*Solanum tuberosum* L.) plants. We specifically investigated the coordinated response of key antioxidant enzymes during co-infection by the hemibiotrophic oomycete *Phytophthora infestans* (Mont.) de Bary and the necrotrophic fungus *Alternaria solani* Sorauer. Our aim was to decipher whether minimal AgNP applications can effectively “hijack” the ROS signaling apparatus to prime a more robust and effective immune response against complex phytopathogenic threats.

## 2. Materials and Methods

Polyhexamethylene biguanide hydrochloride (20 wt. % aqueous solution, Arch Chemicals Inc., Castleford, UK), sodium tallow amphopolycarboxyglycinate (30 wt. % aqueous solution containing 10 wt. % NaCl, Nouryon, Amsterdam, The Netherlands), sodium borohydride (98 wt. %, Geel, Belgium), silver nitrate (99 wt. +%, Ural Chemical Reagents Plant, Sverdlovskaya obl., Russia), ammonia (25 wt. % aqueous solution, Sigma-Aldrich, St. Louis, MO, USA), and hydrogen peroxide (30 wt. % aqueous solution, Sigma-Aldrich, St. Louis, MO, USA) were used as received. Distilled water was used for the preparation of all solutions during each experiment.

### 2.1. Synthesis of AgNPs Stabilized with Polymers

Aqueous AgNP dispersions stabilized with sodium tallow amphopolycarboxyglycinate (TAP) [15], and polyhexamethylene biguanide hydrochloride (PHMB) [16] (Figure 1) were obtained via the common technique of AgNO_3_ reduction using sodium borohydride in the presence of a stabilizer [17]. The dispersions were fully characterized earlier [15,16], using conventional methods [18].

A minimal amount of the stabilizer needed to prepare a AgNP dispersion with high colloidal stability and without a tendency for NP aggregation within the period of the first three months was added to the reaction mixture: 500 mg/L of TAP or 100 mg/L of PHMB.

### 2.2. Synthesis of Ag-AgCl Composite Nanoparticles Stabilized with Sodium Tallow Amphopolycarboxyglycinate

Nanocomposites Ag-AgCl stabilized with TAP were synthesized by UV reduction in colloidal silver chloride in the presence of hydrogen peroxide and 500 g/L of sodium tallow amphopolycarboxyglycinate [19].

### 2.3. Registration of Absorption Light-Spectra

To register absorption spectra in the visible region of 330–700 nm using quartz cuvettes (optical path length 10 mm), a spectrophotometer UV-1800 (Shimadzu, Kyoto, Japan) was used.

### 2.4. Transmission and Scanning Electron Microscopy with Electron Probe Micro-Analyzer (EPMA)

Micrographs of specimens of nanoparticles were made on a transmission electron microscope LEO 912 AB OMEGA (Carl Zeiss, Oberkochen, Germany) operating at 100 kV and a scanning electron microscope (Carl Zeiss NVision 40, equipped with Oxford Instruments x-Max energy dispersive X-ray detector). The specimens were prepared by placing 1–2 μL of the dispersion on a Formvar™-coated copper grid, which was then dried in air.

### 2.5. DLS

The hydrodynamic diameters and ζ-potential of the obtained NPs were determined using a Zetasizer Nano ZS (Malvern Instruments, Malvern, UK). Measurements were performed in quartz cuvettes (10 mm optical path; scattering angle 90°) at a concentration of AgNPs of 50 mg/L. The data were processed in DynaLS v 2.0 software using a monomodal Gaussian distribution model.

### 2.6. X-Ray Analysis

Diffraction patterns were recorded at the “Structural Materials Science” end-station at the Kurchatov synchrotron radiation source. For our measurements, we used an SI beam which was passed through a single crystal monochromator with a cut of Si (2 2 0) and was pre-configured to the Zr K-edge (λ = 0.688 86 Å) based on the intensity of the radiation absorption in the zirconium foil. The beam was limited to 300 × 300 μm in size by the collimator shutters. We used a photographic plate, Fuji ImagingPlate, as a detector and fixed it in a special plane holder at a distance of 200 mm from the sample. To define the distance between the sample, the detector, and the angle between the beam direction and the plane of the detector more accurately, a powder-like standard Si certified by the National Institute of Standards and Technology (NIST, Gaithersburg, MD, USA) was used. To convert 2D diffraction patterns into 1D dependences I(2θ), the program Fit2d V12.012 was used.

### 2.7. Determining Activity of Peroxidase

The activity of POX was determined by means of a common spectrophotometric technique [20], based on the assessment of the velocity of the chemical reaction of benzidine (4,4′-diaminobiphenyl) oxidation in the presence of hydrogen peroxide and POX.

A total of 200–300 mg of raw leaves was ground in a cold porcelain mortar with a cold pestle with 0.5 mL of acetate buffer (pH 5.0). The obtained homogenate was centrifuged for 5 min at 12,000× *g* and cooled down to 4 °C.

A total of 0.98 mL of 0.2 M acetate buffer (pH 5.0), 0.5 mL of a 100 mg∙L^−1^ solution of benzidine hydrochloride, 0.02 mL of leaf extract, and 0.5 mL of 3 g/L of hydrogen peroxide were put into the spectrophotometer cuvette. In total, 1.48 mL of a 0.2 M acetate buffer (pH 5.0), 0.5 mL of a 100 mg/L solution of benzidine hydrochloride, and 0.02 mL of the leaf extract were put into the control cuvette. Optical density was measured at a wavelength of 590 nm every second for 120 s.

The activity of POX (per gram of the dry weight of the leaves) was calculated in relative units. The activity of POX was measured in three biological and analytical repetitions.

### 2.8. Determining Activity of Catalase

The activity of catalase was determined by means of the spectrophotometric technique [20], as in the assessment of the velocity of the chemical reaction of the decomposition of hydrogen peroxide with catalase.

A total of 250 mg of raw leaves was ground in a cold porcelain mortar with 0.5 mL of extraction buffer solution (50 mM phosphate buffer, pH = 7.0). The obtained homogenate was centrifuged for 5 min at 12,000× *g* and cooled down to 4 °C. Next, 2.95 mL of 50 mM phosphate buffer (pH = 7.0), 30 µL of extract, and 20 µL of 0.6 M hydrogen peroxide were put into the spectrophotometer cuvette. Finally, 2.95 mL of 50 mM phosphate buffer (pH = 7.0) and 30 µL of extract were put into the control cuvette. Optical density was measured at a wavelength of 240 nm every second for 120 s.

The activity of catalase in relative units per one gram of the dry weight was determined. The activity of catalase was measured in three biological and analytical repetitions.

### 2.9. Field Trials

In field trials, aqueous AgNP dispersions stabilized with PHMB and TAP, as well as an Ag-AgCl nanocomposite stabilized with TAP, were used. Field trials were conducted from June to September 2024 in Tambov Oblast, Russian Federation (52° N, 41° E), on leached chernozem (Haplic Chernozem, FAO classification). The experimental site had been planted with spring barley in the previous vegetation season. The potato cultivar *Solanum tuberosum* L. ‘Nevsky’ (early–mid maturity group) was planted using first-generation mini-tubers (50–60 g average weight). The experiment followed a randomized complete block design with four replicates. Each plot measured 100 m^2^ (10 m × 10 m) and was separated by 1 m buffer zones to prevent cross-contamination.

All plots received a baseline NPK fertilization of 20 kg/ha each of nitrogen (as ammonium nitrate), phosphorus (as diammonium phosphate), and potassium (as potassium chloride), applied uniformly during pre-plant soil preparation. Foliar treatments were applied using a calibrated backpack sprayer equipped with flat-fan nozzles (XR 11002 Tecomec S.r.l., Reggio Emilia, Italy), delivering a consistent spray volume of 200 L/ha. Applications were performed in the evening (18:00–20:00) under calm, dew-free conditions at 18–21 °C. Treatments were administered at three key phenological stages: budding (BBCH 55), canopy closure (BBCH 75), and 10 days thereafter.

The treatments were as follows:Control: background fertilization NPK 20:20:20 kg/ha;NPK + PHMB: three foliar applications of PHMB 0.5 g/L—200 mL/ha: budding (BBCH 55), canopy closure (BBCH 75), and 10 days thereafter;NPK + TAP: three foliar applications of TAP 48 g/L—3 L/ha: budding (BBCH 55), canopy closure (BBCH 75), and 10 days thereafter;NPK + AgNPs-PHMB: three foliar applications of AgNPs—0.5 g/L Ag—stabilized with PHMB 0.5 g/L—200 mL/ha, Ag input: 0.1 g/ha: budding (BBCH 55), canopy closure (BBCH 75), and 10 days thereafter;NPK + AgNPs-TAP: three foliar applications of AgNPs—3 g/L Ag—stabilized with TAP 48 g∙L^−1^—3 L/ha, Ag input: 9.0 g/ha: budding (BBCH 55), canopy closure (BBCH 75), and 10 days thereafter;NPK + Ag-AgCl/TAP: three foliar applications of Ag-AgCl nanocomposite dispersion 2.5 g/L Ag, 48 g/L TAP—3 L/ha, Ag input: 7.5 g/ha: budding (BBCH 55), canopy closure (BBCH 75), and 10 days thereafter;NPK + Ag-AgCl/TAP: three foliar applications of Ag-AgCl nanocomposite dispersion 2.5 g/L Ag, 48 g/L TAP—250 mL/ha, Ag input: 0.6 g/ha: budding (BBCH 55), canopy closure (BBCH 75), and 10 days thereafter.

Weeds were controlled uniformly across all plots using the standard herbicide program of the farm, with applications at the pre-emergence and canopy closure stages (BBCH 75).

All spray solutions were prepared with deionized water adjusted to pH 6.5 ± 0.2 and an electrical conductivity of 0.8 dS/m. Nanodispersions were sonicated for 10 min before application to ensure homogeneity.

Disease assessment was conducted weekly from canopy closure until senescence. Late blight (*Phytophthora infestans*) severity was scored visually on leaves using a 0–9 scale (0 = no symptoms, 9 = >90% leaf area affected). I = The incidence (%) was calculated as *P* = *n*/*N*, where *n* is the number of symptomatic plants, and *N* is the total number assessed (n = 50 per plot). Disease control efficacy (%) was calculated using Abbott’s formula: *E* = (*Pc* − *Pt*)/*Pc* × 100, where *Pc* and *Pt* are disease incidence in the control and treated plots, respectively.

Yield and tuber grading: At harvest, tubers were manually collected from each plot, washed, and sorted into marketable fractions: small (<30 mm), seed (30–60 mm), and ware (>60 mm). Total and fraction yields were recorded in t/ha.

Leaf sampling for biochemical assays: Fully expanded leaves were collected at canopy closure (prior to the second treatment) and at flowering (after the second treatment). Samples were immediately flash-frozen in liquid nitrogen and stored in Dewar flasks at −196 °C until analysis of antioxidant enzyme activities.

Statistical analysis: The normality of the data was assessed using the Shapiro–Wilk test, and the results confirmed that the data were normally distributed (*p* > 0.05). Data were analyzed by one-way ANOVA followed by Fisher’s LSD test (*p* < 0.05) using Statistica 12.0. Experimental design and disease evaluation followed EPPO standard PM 7/98 [21].

## 3. Results and Discussion

### 3.1. Synthesis, Physical, and Chemical Properties of AgNPs

All AgNP dispersions used in this study were synthesized and characterized according to our previously published protocols. The resulting colloidal suspensions were brightly colored liquids, exhibiting a distinct surface plasmon resonance (SPR) absorption band typical of AgNPs in their UV-Vis spectra (Figure 2), confirming the formation of nanoscale metallic particles.

The presence of nanoparticles was further confirmed by transmission electron microscopy (TEM) (Figure 3). Particle size analysis revealed a relatively narrow size distribution in all dispersions (Figure 4, Table 1), indicating high homogeneity. It should be noted that smaller nanoparticles are generally associated with enhanced biological activity due to their higher specific surface area and improved cellular uptake potential. Dynamic light scattering (DLS) showed hydrodynamic diameters consistent with TEM observations, with zeta potentials exceeding ±30 mV (Table 1), indicating good colloidal stability.

X-ray diffraction (XRD) analysis confirmed the crystalline nature of silver in the AgNPs stabilized with PHMB and TAP (Figure 5). In the Ag-AgCl nanocomposite dispersion, TEM and XRD data indicate that small metallic silver nanoparticles were embedded within larger silver chloride (AgCl) crystalline matrices, suggesting a core–shell or dispersed-phase structure.

Dispersions remained stable for over 6 months without aggregation, as confirmed by unchanged SPR peaks and particle size distribution.

### 3.2. Field Trials Results

Foliar applications of AgNP dispersions had no significant effect on potato emergence, duration of the vegetative period, or timing of key phenological stages. Uniform emergence was observed across all experimental variants, including the control, with a final emergence rate of 100%.

During the growing season, natural infection caused by *A. solani* (causing early blight) and *P. infestans* (causing late blight) was recorded on foliage starting from the budding stage and progressing throughout the summer (Figure 6, Table 2). In the control plots, disease incidence remained low (<10%) during early vegetation but gradually increased in SOD, reaching moderate severity for *A. solani* and high severity for *P. infestans* by mid- to late season.

All AgNP treatments effectively suppressed both diseases, providing disease control efficacy of at least 60%. Notably, the dispersions exhibited particularly strong suppression of *A. solani*, with a success rate exceeding 80%. In contrast, the individual stabilizing agents—added to prevent nanoparticle aggregation—showed statistically significant (*p* < 0.05, ANOVA) 10–30% lower antifungal activity compared to the complete silver formulations, indicating that the primary protective effect is attributable to AgNPs rather than the stabilizers alone. The key role of stabilizing agents is to maintain aggregative stability of AgNP colloids during foliar treatment on potato leaves and inside the plant tissues. Also, the reduction in disease severity was statistically significant in all silver-treated plots compared to the control.

It is well established [22,23,24] that AgNPs exhibit negligible intrinsic fungicidal activity at the concentrations used in the field trials. For example, in a study by Kim et al. [24], silver nanoparticles suppressed the growth of pathogenic fungi by half starting at a concentration of about 10 ppm. In the present study, the silver dose was varied from 0.1 to 9 g/ha, which corresponded to a silver content in the spraying liquid of 0.5–45 ppm, and the silver concentration in the tissues will be significantly lower. Even at a silver content of 0.5 ppm, the disease control efficacy in all cases exceeded 60% and showed little dependence on the applied Ag input across the range of 0.1 to 9.0 g/ha (Table 2), suggesting that, under our field conditions, AgNPs primarily acted by eliciting plant defense responses rather than by exerting direct fungicidal activity against the pathogen.

Therefore, the observed significant suppression of phytopathogens is likely attributable to an indirect, host-mediated mechanism rather than direct antimicrobial action.

Previous studies have demonstrated that both silver ions (Ag^+^) and AgNPs significantly impact the plant antioxidant system [14]. In a study by Bagherzadeh, Homaee, and Ehsanpour [25], it was discovered that both Ag^+^ and AgNPs increase total ROS content, superoxide anions, activities of SOD, CAT, ascorbate peroxidase, and glutathione reductase (GR).

Upon entering plant tissues, Ag^+^ ions primarily deplete the cellular pool of reduced GSH, a crucial low-molecular-weight antioxidant involved in redox homeostasis. In higher plants, detoxification of Ag^+^, Cu^2+^, and other heavy metals involves two major pathways: metallothioneins (gene-encoded cysteine-rich proteins) and phytochelatins—oligopeptides enzymatically synthesized from GSH by phytochelatin synthase [26,27,28].

The diversion of GSH toward phytochelatin biosynthesis compromises its availability for ROS scavenging. Glutathione plays a key role in hydrogen peroxide detoxification via the ascorbate–glutathione cycle, where it acts as a reductant for dehydroascorbate reductase (DHAR), enabling the regeneration of ascorbate. The resulting glutathione disulfide (GSSG) is reduced back to GSH by glutathione reductase (GR). Under Ag^+^ stress, excessive consumption of GSH disrupts this cycle, leading to H_2_O_2_ accumulation and the induction of oxidative stress [10,11,26,27,28,29,30,31].

In response, plants modulate the activity of key antioxidant enzymes. CAT, ascorbate peroxidase (APX), and guaiacol peroxidase (GPX) are widely recognized markers of oxidative stress, with their activity altered following challenges of the biotic and abiotic kind [9].

Notably, in vitro studies show that Ag^+^ and AgNPs inhibit CAT activity [32], likely due to binding to thiol groups or disruption of the heme moiety, while exerting minimal effects on SOD [32].

Peroxidase activity is biphasically regulated: it is promoted at low Ag concentrations (<1 mM) [33,34], but inhibited at higher doses [33,34], suggesting a hormetic response.

When applied foliarly via aqueous dispersions of functionalized AgNPs that release Ag^+^ gradually and at low levels, the resulting oxidative stress remains mild and transient. This suboptimal redox perturbation can act as a priming signal, enhancing plant immunity. ROS function not only as direct antimicrobial agents but also as secondary messengers in defense signaling pathways, including the activation of pathogenesis-related (PR) genes and systemic acquired resistance (SAR).

In contrast, high-dose exposure—particularly through root uptake or direct application of silver salts—overwhelms antioxidant defenses, leading to phytotoxicity, growth inhibition, and metabolic dysfunction.

Importantly, controlled oxidative stress contributes to the initiation of the hypersensitive-like response, which restricts pathogen colonization. The cytotoxicity of ROS toward phytopathogens, combined with localized cell death, enhances disease resistance in plants capable of maintaining physiological balance. Thus, the targeted modulation of oxidative stress by biocompatible AgNPs may represent a promising strategy to prime plant immunity against pathogens, provided the dose and delivery system ensure minimal phytotoxicity.

In all experimental conditions, AgNPs acted as elicitors—i.e., they triggered plant defense responses upon entry into plant tissues, including a transient burst of ROS. This response is consistent with the induction of oxidative stress as a signal for defense activation rather than direct pathogen toxicity.

Measurements of POX and CAT activities in potato leaves revealed a significant increase in both enzymes following treatment with AgNPs (Table 3 and Table 4), indicating the onset of a controlled oxidative stress. This response correlates strongly with the observed suppression of *P. infestans* and *A. solani*: enzyme activity levels were positively correlated with disease control efficacy (Figure 7 and Figure 8), showing Pearson correlation coefficients of R^2^ ≈ 0.8–0.9.

This differential modulation—increased POX activity coupled with partial suppression of CAT—is distinct from the coordinated upregulation of both enzymes previously observed in healthy potato plants of the early cultivar ‘Red Scarlet’ following foliar application of PHMB-stabilized AgNPs in the Pre-Kama region of Tatarstan [35]. A similar pattern has been reported by observing wheat seeds treated with AgNPs under NaCl-induced salt stress [36], though the underlying mechanism was not explained in that study.

Based on our collective findings, we proposed a model of the action of AgNPs with respect to the plant antioxidant system under phytopathogenic stress (Figure 9).

The stabilizing agents alone (PHMB and TAP) exerted considerably weaker effects on antioxidant enzyme activity, suggesting that the primary role in immune activation is played by AgNPs and/or Ag^+^ ions released through oxidation of Ag^0^ in planta under the influence of ROS and apoplastic oxygen.

The activity of catalase and peroxidase in plants treated with stabilizers alone was, in all cases, 15–40% lower than in plants treated with silver dispersions. The only exception refers to the weak influence of PHMB on catalase activity in the case of early blight-infected plants. This further supports the previously obtained evidence for the primary contribution of silver nanoparticles, rather than stabilizers, to preventing potato infections.

Krutyakov et al. [48] previously showed that in the presence of H_2_O_2_, AgNPs stabilized by various surfactants and polymers, including the AgNPs used in the field studies, are readily oxidized by H_2_O_2_ with the release of Ag^+^ ions; the kinetic characteristics of the process were also measured. This suggests that Ag^+^ ions are formed from AgNPs in plant tissues under the influence of H_2_O_2_ and other ROS contained therein.

This upregulation of antioxidant enzymes—accompanied by increased H_2_O_2_ and lipid peroxidation products such as malondialdehyde (MDA)—has been previously reported in plants exposed to AgNPs [36] and supports the proposed mechanism of POX and catalase induction via silver-triggered redox signaling.

To elucidate the mechanism by which AgNPs influence disease development, POX and CAT activities were measured separately in potato leaves exhibiting visible infection foci (Table 3 and Table 4).

In control plants, antioxidant enzyme activity was significantly reduced in infected leaves compared to healthy tissues. Given the extensive pathogen invasion observed during field trials, this decline likely reflects depletion of the plant’s defense resources, contributing to low disease resistance.

In contrast, following protective priming with AgNPs, leaf infection by *A. solani* and *P. infestans* led to a further increase in the POX activity—exceeding levels in healthy, non-inoculated plants. CAT activity decreased by 10–20% upon infection, both in control and AgNP-treated plants, consistent with its known downregulation under biotic stress. However, CAT activity remained significantly higher in AgNP-treated plants compared to the control, indicating a silver-mediated enhancement of enzymatic capacity despite infection.

Upon pathogen recognition, plants may activate a two-tiered immune system [37,38,39,40,41,42] (Figure 9(1)): PAMP-triggered immunity (PTI) and effector-triggered immunity (ETI). PAMPs, such as bacterial flagellin or fungal chitin, are recognized (Figure 9(1)) by membrane-localized receptor-like kinases (LRR-RLKs; e.g., FLS2, EFR, CERK1), triggering early defenses including cell wall reinforcement (callose, lignin), Ca^2+^ uptake (Figure 9(2)), ROS production via plasma membrane NADPH oxidases (RBOHD/RBOHF), activated by Ca^2+^, phytoalexin synthesis, and the activation of pathogenesis-related (PR) genes (Figure 9(3)).

Pathogens counteract PTI by delivering effector proteins into host cells. In pathogen-resistant plants, these effectors are detected directly or indirectly by intracellular nucleotide-binding leucine-rich repeat (NLR) receptors (Figure 9(1)), activating ETI—a rapid, high-amplitude defense response often culminating in hypersensitive cell death at the site of infection [43,44,45]. Signaling downstream of pathogen perception involves Ca^2+^ influx (Figure 9(2)), ROS bursts, MAPK cascades, and hormonal regulation via salicylic acid (SA) for biotroph defense, and jasmonic acid (JA)/ethylene (ET) for necrotroph resistance.

H_2_O_2_ and other ROS alter cellular redox status, promoting monomerization and nuclear translocation of the nonexpressor of pathogenesis-related genes (NPR1) [46] (Figure 9(3))—the central regulator of SAR—and stimulating SA biosynthesis (Figure 9(4)) via isochorismate synthase 1 (ICS1).

SA, in turn, enhances hydrogen peroxide (H_2_O_2_) production, thereby establishing a positive feedback loop that amplifies oxidative stress [47]. Furthermore, SA antagonizes JA-mediated signaling (Figure 9(5)). NPR1, a central regulator of the SA pathway, translocates to the nucleus where it interacts with TGA transcription factors to activate pathogenesis-related (*PR*) genes (Figure 9(14)). Concurrently, NPR1-mediated activation suppresses the expression of JA-responsive genes, such as *LIPOXYGENASE 2* (*LOX2*) and *VEGETATIVE STORAGE PROTEIN 2* (*VSP2*). SA also inhibits JA biosynthesis by repressing the activity of lipoxygenase (LOX), the first enzyme in the JA biosynthetic pathway, and by stabilizing JASMONATE ZIM-DOMAIN (JAZ) repressor proteins [45,47].

Conversely, JA and ET suppress SA signaling. JAZ repressors physically interact with NPR1 and TGA transcription factors, thereby inhibiting SA-dependent gene expression. Ethylene can potentiate this suppression through the transcription factors EIN3 and EIL1 [45,47].

Following foliar application, AgNPs readily bind to cell wall components—particularly phenolic groups in lignin and carboxyl groups in pectins (Figure 9(6)). In the apoplast, dissolved O_2_ and existing ROS promote oxidative dissolution of Ag^0^ to Ag^+^ ions (Figure 9(7)). Ag^+^ enters cells (Figure 9(8)) primarily via copper transporters (COPTs), due to its chemical similarity to Cu^+^. Intracellular detoxification occurs through binding to thiol-containing compounds (Figure 9(9)): GSH (forming Ag^+^(GS)_2_), phytochelatins (PCs, synthesized by PCS), and metallothioneins (MTs). These complexes are sequestered into the vacuole via ABC transporters (Figure 9(10)), while excess Ag^+^ may be extruded or compartmentalized by HMA-family transporters (Figure 9(10)) [13,14,26,27,28].

Two key mechanisms mediate AgNPs’ effect on the cellular metabolism:Induction of oxidative stress via GSH (Figure 9(9)) depletion (diverted to PC synthesis) and disruption of the ascorbate–glutathione cycle, leading to H_2_O_2_ accumulation (Figure 9(11), also via disrupting energy metabolism [49,50];Suppression of ET signaling, potentially through direct interaction with ET receptors (Figure 9(12)).

Based on data reported by Azhar et al. [51], plant cell membranes contain approximately 150 fmol of ethylene receptors per µg of membrane protein, which corresponds to roughly 150 fmol per mg of fresh leaf tissue. Given that the average fresh aboveground biomass of a potato plant at a stand density of 37,000 plants per hectare is approximately 400 g, each plant contains about 60 nmol of ethylene receptors. In field trials, potato plants were treated with nanoparticle dispersions at application rates of 0.1–9 g Ag per hectare, equivalent to 3–240 µg or 30–2200 nmol of silver per plant. Consequently, the amount of applied silver is sufficient to potentially inactivate a substantial fraction of native ethylene receptors.

As previously demonstrated, suppression of the ET signaling pathway results in inhibition of the JA-dependent signaling cascade, thereby promoting activation of the antagonistic SA pathway. A comparable shift toward SA-mediated signaling is also induced by the accumulation of hydrogen peroxide (H_2_O_2_) and other ROS. Consequently, upon intracellular uptake of silver, both mechanisms—ET pathway suppression and ROS accumulation—converge to potentiate SA-dependent immune responses. This, in turn, triggers a robust oxidative burst and elicits a defense reaction phenotypically resembling the hypersensitive response (HR) (Figure 9(13)).

SA signaling leads to transcriptional repression of *CAT* genes (*CAT1*, *CAT2*, *CAT3*) via WRKY transcription factors and oxidative inactivation of CAT enzyme, reducing H_2_O_2_ scavenging. In contrast, POX activity increases via NPR1-dependent expression of class III POX genes (*PRX*). Thus, the observed enzyme dynamics reflect elicitor-driven activation of SA-mediated immunity, when H_2_O_2_ is used for defense instead of being removed.

In this context, the alterations in CAT and POX activities observed in our experiments are primarily attributable to the elicitor-like effects of AgNPs and Ag^+^ ions. Upon interaction with components of the plant cell wall, these agents stimulate the activity of both plasma membrane-localized NADPH oxidases (RBOHs) and cell wall-associated POXs, leading to a concomitant increase in apoplastic and symplastic ROS levels. This ROS burst acts as a secondary messenger, triggering the activation of transcription factors and subsequent upregulation of defense-related genes, including canonical resistance (*R*) genes.

Consequently, the plant prioritizes the synthesis of antimicrobial compounds and defense enzymes directly involved in pathogen containment—particularly ROS, POXs, and polyphenol oxidases—while catalase, which scavenges protective ROS, is produced in lesser quantities. This shift in the ROS-scavenging versus ROS-generating enzyme balance reflects a strategic reallocation of cellular resources toward defense.

Pre-treatment with AgNPs thus induces a state of defense priming: upon subsequent pathogen encounter, plants mount a rapid and robust non-specific immune response (Figure 10). This primed state effectively restricts pathogen progression and prevents the exhaustion of the plant’s defensive capacity during prolonged infection.

Furthermore, considering recent evidence demonstrating the direct effects of silver ions and AgNPs on the catalytic activity of antioxidant enzymes [32,33,34], an alternative mechanistic explanation for the observed enzymatic shifts can be proposed. Specifically, the increase in POX activity coupled with the decline in CAT activity may be caused not only by transcriptional regulation but also by the direct interaction of Ag^+^ and AgNPs with enzyme molecules. Studies have shown that, within a specific concentration range, Ag^+^ can enhance POX activity while simultaneously inhibiting CAT [32,33,34].

Silver ions are released in plant tissues through the oxidative dissolution of AgNPs, a process accelerated by the increase in the concentration of oxidants such as molecular oxygen or ROS. Under conditions of mild oxidative stress—as typically observed in healthy, uninfected plants—the rate of AgNPs dissolution is limited, resulting in low Ag^+^ concentrations that exert negligible direct effects on POX or CAT activity. However, during co-exposure to phytopathogens and AgNPs, oxidative stress is markedly amplified compared to the non-infected control. This leads to a substantial ROS burst, which in turn accelerates AgNPs dissolution and promotes localized accumulation of Ag^+^ in infected tissues. Under these conditions, the direct modulatory effects of Ag^+^ on enzyme conformation and function become significant: POX activity is potentiated, whereas CAT activity is suppressed—consistent with in vitro observations of Ag^+^-enzyme interactions [32,33,34].

Foliar application of AgNPs to potato plants thus can enhance disease resistance through a dual mechanism: (i) AgNP-induced ROS generation primes defense signaling, and (ii) the consequent rise in POX activity—potentiated both transcriptionally and post-translationally by Ag^+^—amplifies antimicrobial responses. Notably, despite the perturbation of redox homeostasis, treated plants maintain sufficient antioxidant capacity, as evidenced by their ability to upregulate both POX and CAT in response to pathogen invasion. The pivotal role of POXs loosely bound to the cell wall in conferring biotic stress tolerance in potato has been previously established.

Unlike CAT and SOD, whose primary function is ROS scavenging, POX-catalyzed reactions generate products with direct antimicrobial activity. For instance, oxidation of phenolic compounds by H_2_O_2_ in the presence of POX yields quinones and other derivatives exhibiting strong fungicidal properties [6]. Additionally, these reactions produce high-molecular-weight, poorly soluble lignin-like polymers that reinforce cell walls, forming a physical barrier against pathogen spread and contributing to the formation of necrotic lesions—a hallmark of the hypersensitive response in infected leaves and stems [52,53].

Moreover, POXs serve as a major source of apoplastic ROS during pathogenesis. Therefore, AgNP-mediated enhancement of POX activity not only bolsters chemical defense but also sustains the oxidative burst required for effective pathogen containment.

Based on our findings and the established interplay between plant hormonal pathways, we propose the following mechanistic hypothesis to explain the observed peroxidase (POX) increase, catalase (CAT) decrease, and high disease control efficacy. Following foliar application, silver nanoparticles (AgNPs) are oxidized in the apoplast by tissue-derived hydrogen peroxide (H_2_O_2_) and other ROS, releasing Ag^+^ ions (Stage 1). These ions, and potentially the nanoparticles themselves, then activate the plant’s ROS-mediated signaling network (Stage 2). A critical step involves the direct inhibition of membrane-localized ethylene receptors by Ag^+^ (Stage 3). This suppression of ethylene signaling consequently inhibits the antagonistic jasmonic acid (JA) pathway, leading to a shift in hormonal balance towards the salicylic acid (SA)-mediated signaling branch (Stage 4). The potentiated SA pathway triggers a robust oxidative burst, characterized by the strategic reprogramming of antioxidant enzymes: an upregulation of POX (to generate antimicrobial compounds and reinforce cell walls) and a concurrent downregulation of CAT (to preserve signaling H_2_O_2_). This primed, amplified oxidative response (Stage 5) directly and indirectly creates a hostile environment for the pathogens, leading to the significant suppression of *P. infestans* and *A. solani* infections observed in our field trials.

This mechanism is consistent with the outcome of our recent and previous [54] field trials, which demonstrated reduced disease incidence and improved tuber yield in AgNP-treated plots compared to both the untreated control and the plots treated with stabilizers alone (Table 5). Yield improvement was primarily due to the higher proportion of medium-sized tubers and fewer small tubers (Table 5, Figure 11).

In summary, when applied at optimized rates—where the elicitor-like, defense-priming effects of silver prevail over its phytotoxic potential—foliar sprays of AgNP dispersions effectively induce SAR and suppress disease progression. This translates into measurable agronomic benefits, including lower infection severity and increased potato yield.

Importantly, AgNP treatments did not significantly alter dry matter content, starch, vitamin C, or nitrate levels in tubers compared to the control (Table 6), indicating no adverse impact on nutritional quality.

However, the prospective agricultural use of AgNPs as pesticides raises significant environmental concerns regarding their persistence, bioaccumulation potential, and effects on non-target soil ecosystems. The environmental impact of AgNPs is critically dependent on their synthesis route and subsequent transformations in the environment.

Silver nanoparticles do not persist indefinitely in their pristine metallic (Ag^0^) form in soil environments. They undergo a series of dynamic transformations that dictate their long-term fate and bioavailability.

The primary processes include aggregation (clumping of particles), oxidative dissolution (release of toxic Ag^+^ ions), and sulfidation. Sulfidation, where Ag^0^ reacts with sulfide to form Ag_2_S, is particularly significant as it dramatically reduces the solubility and antimicrobial potency of AgNPs. This is the main pathway of inactivating AgNPs in soils [13].

AgNPs can enter and move through both aquatic and terrestrial food chains, posing a risk of biomagnification. Terrestrial plants can absorb AgNPs through roots or foliar application. Once absorbed, nanoparticles can be translocated within the plant system. Evidence confirms the trophic transfer of various nanoparticles. For example, studies show that nanoparticles can be transferred from algae to water fleas (*Daphnia magna*) and further to fish, accumulating in tissues like the liver, kidney, and muscle [55]. This demonstrates a clear pathway for AgNPs to enter and ascend the food chain, with potential implications for higher organisms, including humans.

Modern research reveals a crucial and concerning difference between AgNPs stabilized by different agents regarding a long-term environmental risk. AgNPs stabilized by biodegradable polymers like those used in our field trials often exhibit less environmental risk than other AgNPs.

## 4. Conclusions

Our study highlights functionalized AgNPs as effective priming agents capable of enhancing systemic resistance through the induction of controlled oxidative stress without exerting direct antimicrobial activity. The field trials results demonstrate that triple foliar application of polymer-stabilized silver nanoparticles (AgNPs) at doses as low as 0.1–9.0 g/ha effectively primed potato (*Solanum tuberosum* L.) plants against late blight (*Phytophthora infestans*) and early blight (*Alternaria solani*), achieving a disease control efficacy of more than 60%. This outcome was linked to the well-coordinated reconfiguration of the plant’s antioxidant system, marked by a significant increase in peroxidase activity (by 30–50%) and a concomitant decrease in catalase activity. This specific enzymatic shift indicates a potentiated, pathogen-responsive oxidative burst, the cornerstone of induced immunity. Consequently, AgNPs applied at these low, non-fungicidal concentrations successfully enhanced the plant’s innate defensive capacity without impairing growth or development, ultimately contributing to the measurable improvement in tuber yield. In potato cultivation, AgNP application has been shown to increase the intensity of antioxidant defense reactions, prime potato plants against phytopathogens, and increase their yield potential. These findings underscore the promise of functionalized AgNPs not as traditional biocides, but as intelligent elicitors of plant immunity, positioning them as next-generation tools for sustainable crop protection and yield optimization, which can be applied at extremely low doses of active substance.

The induction of plant resistance through the activation of intrinsic immune mechanisms offers a sustainable alternative to conventional chemical pesticides, which, due to their widespread use, threaten ecosystem integrity, biodiversity, and human health. In this context, targeted stimulation of plant innate immunity emerges as an eco-compatible strategy that harnesses the plant’s own defense systems, aligning with integrated pest management and the United Nations Sustainable Development Goals (SDG 2 and SDG 12).

## Figures and Tables

**Figure 1 plants-15-00203-f001:**
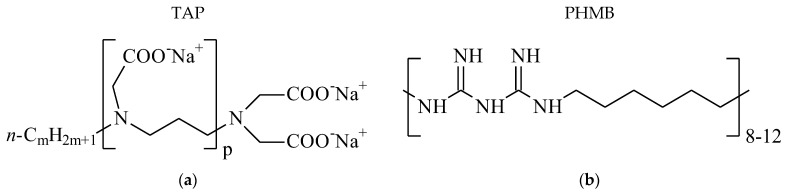
Stabilizers: (**a**) TAP, m = 8–22, p = 2–3, and (**b**) PHMB.

**Figure 2 plants-15-00203-f002:**
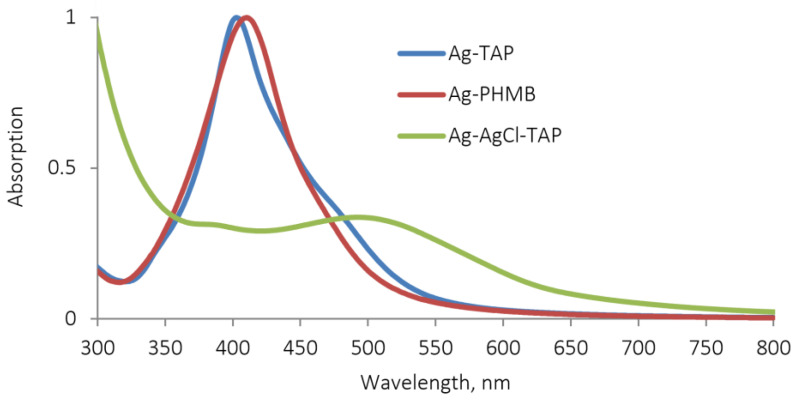
UV-vis adsorption spectra of AgNPs, stabilized with PHMB, TAP, and Ag-AgCl NPs, stabilized by TAP.

**Figure 3 plants-15-00203-f003:**
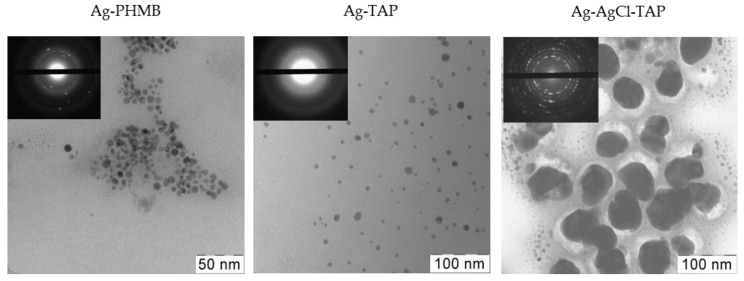
TEM images of AgNPs stabilized with PHMB, TAP, and Ag-AgCl NPs stabilized by TAP.

**Figure 4 plants-15-00203-f004:**
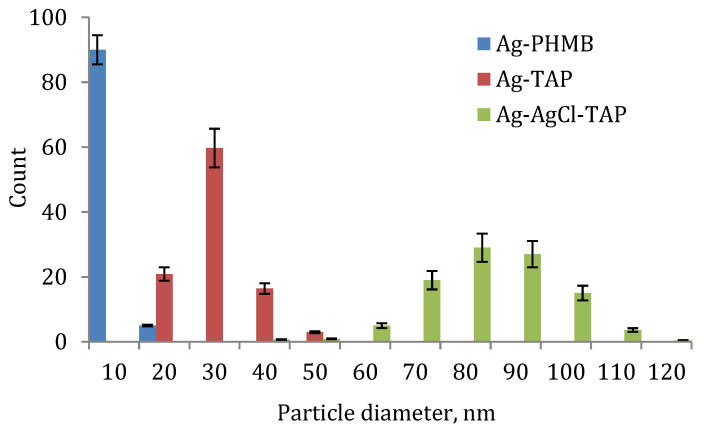
Particle size distribution of Ag NPs, stabilized with PHMB, TAP, and Ag-AgCl NPs, stabilized with TAP.

**Figure 5 plants-15-00203-f005:**
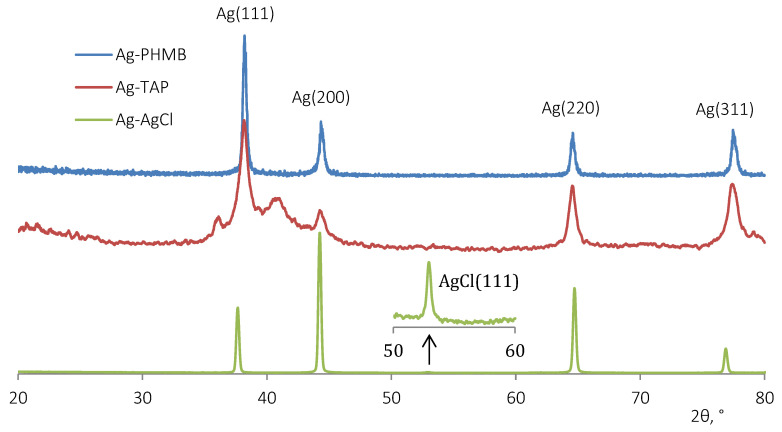
X-ray powder diffraction (XRD) patterns of Ag NPs stabilized with PHMB, TAP, and Ag-AgCl NPs stabilized with TAP.

**Figure 6 plants-15-00203-f006:**
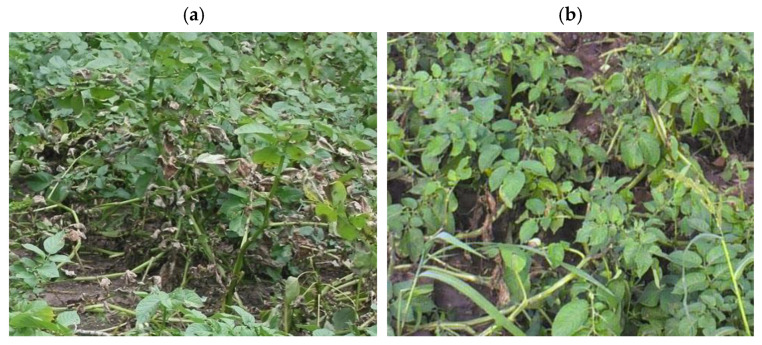
Potato plants infected with late blight, control variant (**a**), and 10 days after first treatment by Ag NPs stabilized with TAP (**b**).

**Figure 7 plants-15-00203-f007:**
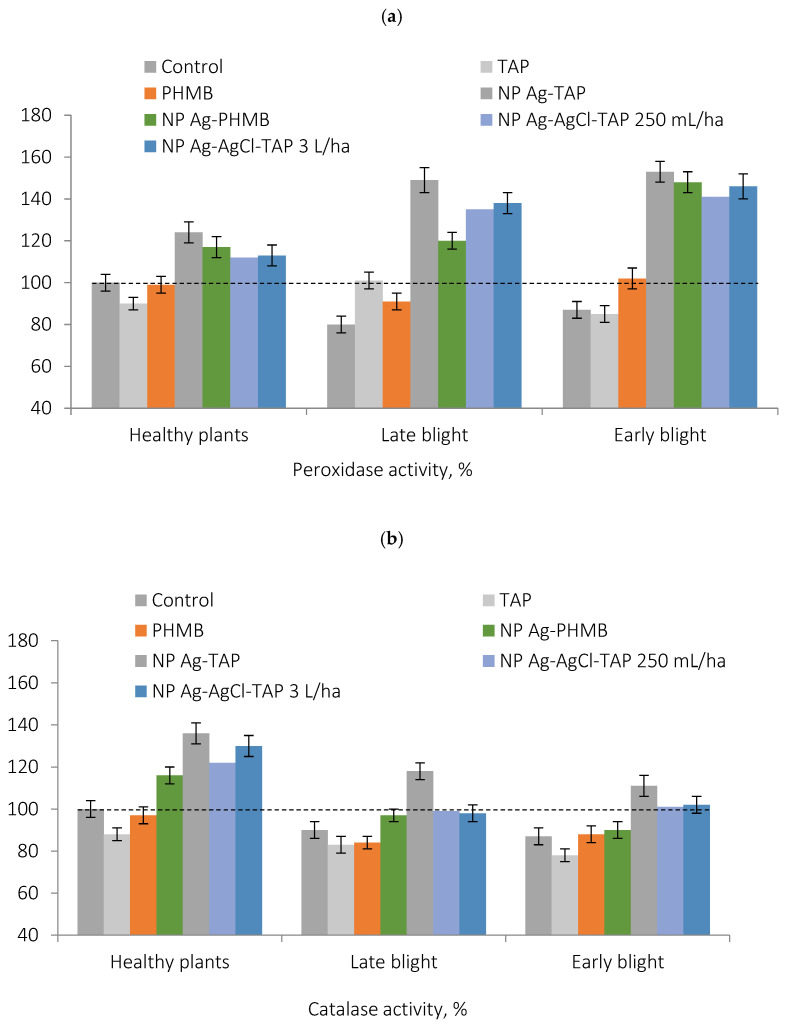
Effect of foliar treatment on peroxidase (**a**) and catalase activity (**b**) in infected and healthy potato leaves, compared to the control, %, relative units.

**Figure 8 plants-15-00203-f008:**
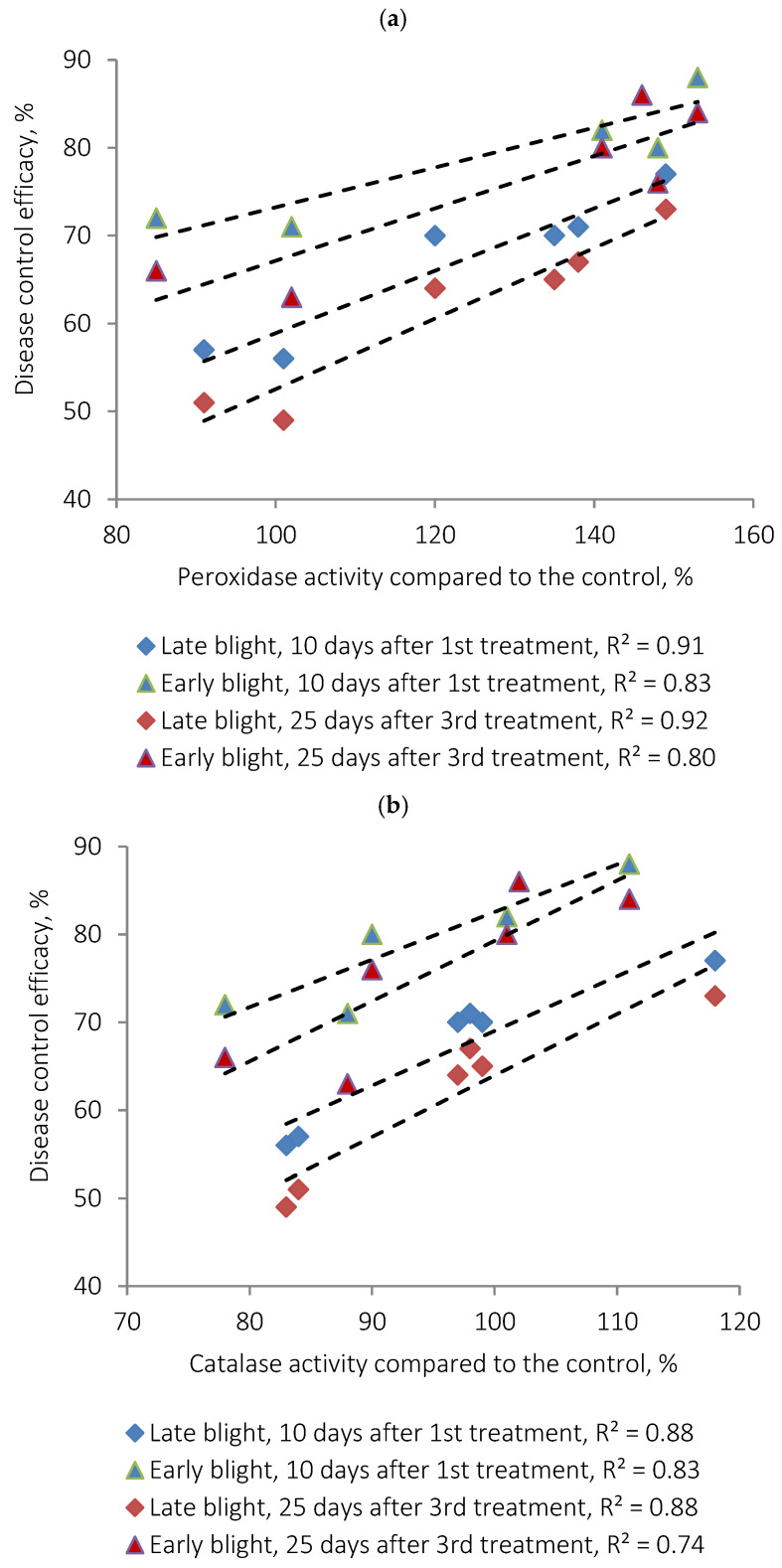
Correlation between peroxidase (**a**) and catalase (**b**) enzymatic activities (relative units) and the efficacy of suppression of late blight and early blight.

**Figure 9 plants-15-00203-f009:**
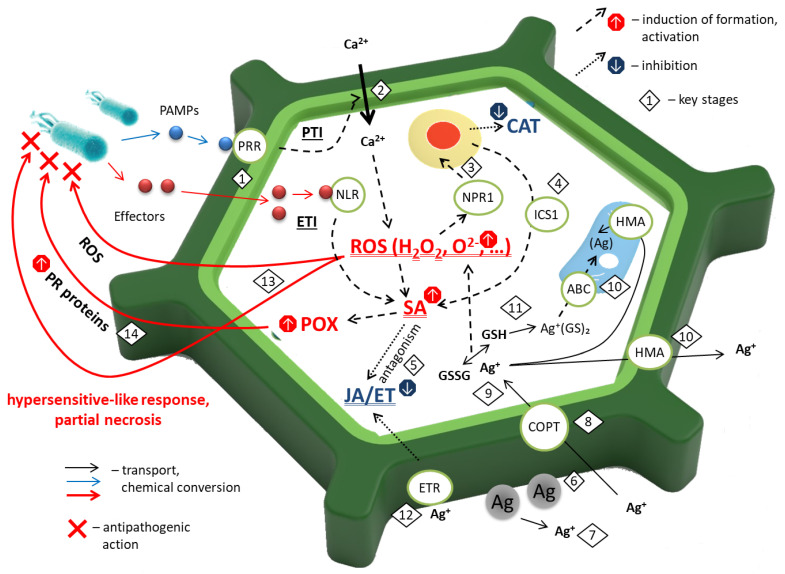
Proposed scheme of the influence of AgNPs on the plant antioxidant system under pathogen infection, summarized from [9,10,11,13,14,25,26,27,28,29,30,31,36,37,38,39,40,41,42,43,44,45,46,47,48]. Key stages: 1—delivering PAMP and effector proteins into host cells; 2—Ca^2+^ uptake induction; 3—activation of pathogenesis-related (PR) genes via NPR1; 4—stimulating of isochorismate synthase 1 (ICS1) production; 5—JA signaling inhibition by SA; 6—AgNPs binding onto cell wall; 7—production of Ag^+^ ions by oxidative dissolution of AgNPs; 8—Ag^+^ influx; 9—Ag^+^ detoxication via binding to GSH, phytochelatins, metallothioneins, resulting in GSH depletion; 10—silver complexes transfer to vacuole and apoplast; 11—induction of ROS formation by GSH deficiency via disruption of the ascorbate-glutathione cycle; 12—ET receptors inhibition by Ag^+^; 13—cell response to pathogen invasion; 14—promoting PR-proteins production by SA.

**Figure 10 plants-15-00203-f010:**
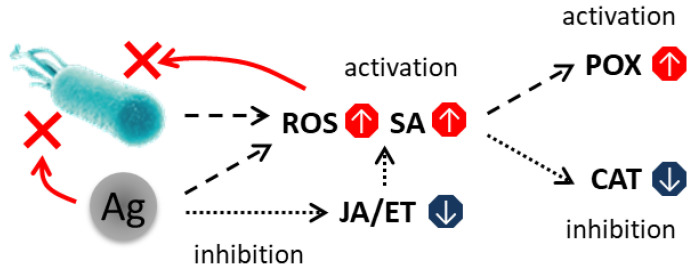
Generalized scheme of the response of AgNP-treated plants to phytopathogen invasion.

**Figure 11 plants-15-00203-f011:**
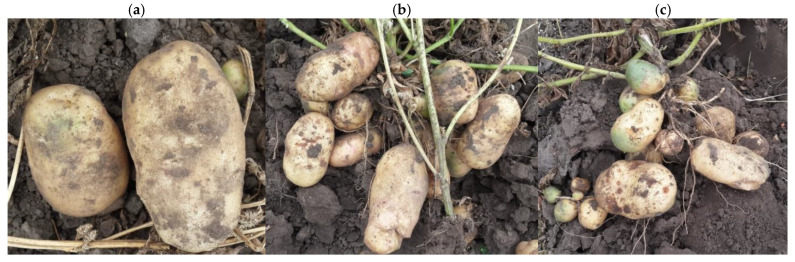
Potato tubers after treatments with (**a**) NP Ag-TAP, (**b**) NP Ag-PHMB, and (**c**) control variant.

**Table 1 plants-15-00203-t001:** Particle size and zeta-potential of Ag NPs stabilized with PHMB, TAP, and Ag-AgCl NPs stabilized with TAP. Standard error of means (*p* = 0.95) is shown.

Colloid	Mean Particle Size, nm	Zeta-Potential
NP Ag-PHMB	10 ± 5	+46 ± 1
NP Ag-TAP	30 ± 5	−54 ± 1
NP Ag-AgCl-TAP	85 ± 5	−35 ± 1

**Table 2 plants-15-00203-t002:** Effect of foliar treatment on the development of late blight and early blight in potato plants. Standard error of means (*p* = 0.95) is shown.

Variant		Ag Input, g/ha	Late Blight		Early Blight	
			10 Days After First Treatment	25 Days After Third Treatment	10 Days After First Treatment	25 Days After Third Treatment
**Disease incidence, %**						
1	Control	–	7.5 ± 0.4	46 ± 2	9.9 ± 0.5	25 ± 1
**Disease control efficacy, %**						
2	PHMB	–	57 ± 4	51 ± 4	71 ± 4	63 ± 4
3	TAP	–	56 ± 3	49 ± 3	72 ± 3	66 ± 4
4	NP Ag-PHMB	0.1	70 ± 4	64 ± 4	80 ± 4	76 ± 3
5	NP Ag-TAP	9.0	77 ± 4	73 ± 4	88 ± 4	84 ± 5
6	NP Ag-AgCl-TAP	7.5	71 ± 5	67 ± 3	86 ± 5	86 ± 4
7	NP Ag-AgCl-TAP	0.6	70 ± 4	65 ± 4	82 ± 4	80 ± 4

**Table 3 plants-15-00203-t003:** Effect of foliar treatment on POX activity in infected and healthy potato leaves compared to the control, %, relative units. Standard error of means (*p* = 0.95) is shown.

Variant		Healthy Plants	Late Blight Infected Plants	Early Blight Infected Plants
1	Control	100	80 ± 4	87 ± 4
2	PHMB	99 ± 5	91 ± 6	102 ± 5
3	TAP	90 ± 4	101 ± 5	85 ± 5
4	NP Ag-PHMB	117 ± 6	120 ± 6	148 ± 7
5	NP Ag-TAP	124 ± 6	149 ± 6	153 ± 7
6	NP Ag-AgCl-TAP 3 L/ha	113 ± 5	138 ± 7	146 ± 6
7	NP Ag-AgCl-TAP 250 mL/ha	112 ± 5	135 ± 6	141 ± 6

**Table 4 plants-15-00203-t004:** Effect of foliar treatment on catalase activity in infected and healthy potato leaves compared to the control, %, relative units. Standard error of means (*p* = 0.95) is shown.

Variant		Healthy Plants	Late Blight Infected Plants	Early Blight Infected Plants
1	Control	100	90 ± 5	87 ± 5
2	PHMB	97 ± 5	84 ± 5	88 ± 4
3	TAP	88 ± 5	83 ± 4	78 ± 4
4	NP Ag-PHMB	116 ± 7	97 ± 6	90 ± 6
5	NP Ag-TAP	136 ± 6	118 ± 6	111 ± 7
6	NP Ag-AgCl-TAP 3 L/ha	130 ± 7	98 ± 5	102 ± 5
7	NP Ag-AgCl-TAP 250 mL/ha	122 ± 6	99 ± 5	101 ± 5

**Table 5 plants-15-00203-t005:** Potato yield: effect of foliar application of AgNPs on yield structure.

Variant		Total Yield, t/ha	Yield Increase Compared to the Control, %	Proportion of the Specified Size Fraction, %		
				>60 mm	30–60 mm	˂30 mm
1	Control	36.3	–	22.6	40.4	37.0
2	PHMB	37.1	2.2	25.0	42.7	32.3
3	TAP	36.9	1.7	24.7	43.1	32.2
4	NP Ag-PHMB	38.1	5.0	25.5	48.2	26.3
5	NP Ag-TAP	40.7	12.1	28.6	56.1	15.3
6	NP Ag-AgCl-TAP	40.0	10.2	26.1	53.8	20.1

**Table 6 plants-15-00203-t006:** Biochemical quality parameters of potato tubers.

Variant		Starch, %	Dry Matter, %	Ascorbic Acid, ppm	Nitrates, mg/kg
1	Control	14.2	18.2	138	170
2	PHMB	14.2	18.4	140	170
3	TAP	14.3	18.5	145	180
4	NP Ag-PHMB	14.6	18.5	144	190
5	NP Ag-TAP	14.8	18.7	146	170
6	NP Ag-AgCl-TAP	14.6	18.6	147	170

## Data Availability

The original contributions presented in this study are included in the article. Further inquiries can be directed to the corresponding author.

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
