# Peer review of "Surface-Functionalized Silver Nanoparticles Boost Oxidative Stress and Prime Potatoes Against Phytopathogens"

_plants, 2026, doi:10.3390/plants15020203_

Round 1

Reviewer 1 Report

Comments and Suggestions for Authors

This manuscript reports the preparation, physicochemical characterization, and biological evaluation of silver-based nanoparticle formulations designed to enhance oxidative stress responses and disease resistance in potato plants (Solanum tuberosum L.). The study integrates synthetic methodologies, detailed spectroscopic and microscopic analyses, and field-scale biological assessments to elucidate how polymer-stabilized AgNPs modulate antioxidant enzyme activities and prime plant immunity against Phytophthora infestans and Alternaria solani. The combination of laboratory characterization and in vivo validation provides a coherent framework supporting the proposed mechanism of nanoparticle-induced defense priming. However, additional experimental evidence or well-cited literature support is required to strengthen the validity of certain claims. Several critical issues must be addressed through major revisions before the manuscript can be considered for publication.

  1. It is essential to clarify whether the observed improvement in disease control efficacy arises from the direct bactericidal activity of silver (Ag), the induction of reactive oxygen species (ROS) in plants, or a synergistic effect of both mechanisms.
  2. The potential role of the polymer coating material in triggering ROS generation requires further discussion and clarification. A more thorough analysis should be provided to distinguish between nanoparticle core effects and carrier-mediated contributions.
  3. The conclusion that Ag nanoparticles regulate ROS-related enzymes lacks direct experimental support. To substantiate this claim, the authors should either include supplementary data or provide robust references to previously published studies demonstrating similar regulatory effects.
  4. While TEM and XPS analyses confirm the successful encapsulation of Ag, the release behavior of silver from the formulation remains uncharacterized. Information on whether Ag is released under physiological conditions and its approximate release kinetics should be included to better understand the mode of action.
  5. For comprehensive characterization of silver nanomaterials, the authors are encouraged to refer to established methodologies described in the literature (e.g., DOI: 10.1002/smo.20230016) and incorporate relevant analytical approaches where appropriate. This reference should also be cited to support methodological rigor.
  6. If the primary source of ROS is attributed to Ag, the rationale for incorporating the wrapping material should be clearly justified. Specifically, what problem does this formulation solve compared to bare AgNPs? Insights into ROS mechanism from recent studies (e.g., DOI: 10.1016/j.indcrop.2024.119956; 10.1016/j.pestbp.2025.106588) may assist in refining the mechanistic interpretation and should be appropriately referenced.

Author Response

  1. It is essential to clarify whether the observed improvement in disease control efficacy arises from the direct bactericidal activity of silver (Ag), the induction of reactive oxygen species (ROS) in plants, or a synergistic effect of both mechanisms.
    We have clarified the core concept of AgNPs as primers vs. direct biocides, page 9: 290-303.
  2. The potential role of the polymer coating material in triggering ROS generation requires further discussion and clarification. A more thorough analysis should be provided to distinguish between nanoparticle core effects and carrier-mediated contributions.
    We have added discussion on the role of the polymer coating material, including carrier-mediated contributions, page 9: 280-290. 
  3. The conclusion that Ag nanoparticles regulate ROS-related enzymes lacks direct experimental support. To substantiate this claim, the authors should either include supplementary data or provide robust references to previously published studies demonstrating similar regulatory effects.
    We have expanded the literature review on AgNPs and oxidative stress; integrated a clearer, step-wise mechanistic hypothesis linking Ag⁺ release, ethylene receptor inhibition, hormonal pathway crosstalk (SA/JA/ET), and the resulting peroxidase/catalase dynamics, pages 9-10.
  4. While TEM and XPS analyses confirm the successful encapsulation of Ag, the release behavior of silver from the formulation remains uncharacterized. Information on whether Ag is released under physiological conditions and its approximate release kinetics should be included to better understand the mode of action.
    We have cited our previous work on the oxidative dissolution of AgNPs by H₂O₂, providing a physicochemical basis for Ag⁺ release in planta, page 15: 413-418. 
  5. For comprehensive characterization of silver nanomaterials, the authors are encouraged to refer to established methodologies described in the literature (e.g., DOI: 10.1002/smo.20230016) and incorporate relevant analytical approaches where appropriate. This reference should also be cited to support methodological rigor.
    We have added references to our previously published papers detailing the full physicochemical characterization (DLS, SEM, XRD, etc.) of the specific AgNP formulations used and cited DOI: 10.1002/smo.20230016, page 3: 94-96.
  6. If the primary source of ROS is attributed to Ag, the rationale for incorporating the wrapping material should be clearly justified. Specifically, what problem does this formulation solve compared to bare AgNPs? Insights into ROS mechanism from recent studies (e.g., DOI: 10.1016/j.indcrop.2024.119956; 10.1016/j.pestbp.2025.106588) may assist in refining the mechanistic interpretation and should be appropriately referenced.
    We have added discussion on the key role of stabilizers in maintaining colloidal stability during foliar application, on leaf surfaces, and within plant tissues, page 9:280-290.

Reviewer 2 Report

Comments and Suggestions for Authors

The manuscript entitled Harnessing Silver Nanoparticles to Boost Oxidative Stress and 2 Prime Potatoes against Phytopathogens present findings and may appears to be a candidate for publication in this journal. Nevertheless, some critical issues must be addressed prior to acceptance. The authors are requested to very carefully consider (MINOR REVESION) and respond to the following comments:

  1.  The manuscript requires comprehensive proofreading to rectify typographical and grammatical errors.
  2. The initial letters of the keywords should be italicized. The number of keywords should range from 5 to 6, and they should not be duplicated in the title.
  3. Line 67 needs to be properly included.
  4. Attention must be given to correcting technical formatting inconsistencies, such as improper hyphenation and irregular spacing.
  5. Both the abstract and conclusion necessitate substantial revision to enhance clarity and coherence.
  6. The introduction should be more focused and better aligned with the study’s aims and objectives. This section appears disorganized and fails to adequately explain the relevance of the manuscript's title. The sentences should be rephrased, and several typographical errors need to be addressed.
  7. The quality of the images should be improved.
Comments on the Quality of English Language

The English could be improved to more clearly express the research.

Author Response

  1. The manuscript requires comprehensive proofreading to rectify typographical and grammatical errors.
  2. The initial letters of the keywords should be italicized. The number of keywords should range from 5 to 6, and they should not be duplicated in the title.
  3. Line 67 needs to be properly included.
  4. Attention must be given to correcting technical formatting inconsistencies, such as improper hyphenation and irregular spacing.
    1-4: We have corrected the typographical errors, unified terminology, and improved overall formatting.
  5. Both the abstract and conclusion necessitate substantial revision to enhance clarity and coherence.
    We have substantially revised the Abstract, Introduction, and Conclusion.
  6. The introduction should be more focused and better aligned with the study’s aims and objectives. This section appears disorganized and fails to adequately explain the relevance of the manuscript's title. The sentences should be rephrased, and several typographical errors need to be addressed.
    We have completely redesigned Introduction.
  7. The quality of the images should be improved.
    We have comprehensively improved all figures (Graphical Abstract, diagrams, charts) for better clarity and relevance.

Reviewer 3 Report

Comments and Suggestions for Authors

Manuscript ID : plants-4031718 

The manuscript “Harnessing Silver Nanoparticles to Boost Oxidative Stress and Prime Potatoes against Phytopathogens ” presents a study on the roles of silver nanoparticles in enhancing the oxidative stress and improving the tomato immunity against phytopathogens. However, this study has a number of shortcomings. It is suggested to revise the manuscript to better justify the study design and discuss findings in the context of existing studies, as follows. Main questions that should be addressed:

  1. The topic of research is not compelling. Authors should write proper title of this manuscript.
  2. Graphical abstract is very short and unclear. It is suggested to include plant image and highligh the key findings in graphical abstract.
  3. Authors should focus on the key findings of their study.
  4. In abstract, there in no information how treatments were applied. A quantitative description of the percentage changes is missing.
  5. The reader is left with a list of findings rather than a clear understanding of the overall mechanism by which silver nanoparticlessuppress disease severity in tomato plants. Highlight the significance of findings while maintaining scientific rigor.
  6. The closing statement is very weak. The abstract doesn't provide a strong conclusion on the significance of the study.

7. In introduction, the first paragraph is rough and most of the provided information is not necessary. The lack of smooth transitions makes it difficult to follow the main argument

8. Authors should elaborate on the siglaing or defense mechanisms of various nanomaterials in managing the pathogen infestation and then clearly establish the study's objective, demonstrating how this research advances beyond and differs from prior investigations

9. Please ensure that the full names of crops, NPs, and metals are used upon their first mention in the text. Subsequently, abbreviated or short names can be employed.

10. The objective stated is broad and doesn't clearly highlight the novelty or significance of the study. It does not tell the readers what the expectation is, or why this is important. Clearly articulate the hypothesis or research question to guide the reader on the study's focus.

11. Materials and methods. Keep the units same. There are few typo errors in this section. Kindly recheck in whole manuscript.

  1. In materials andmethods, mention in detail which plant part was taken for the determination of assays such as peoxidase and acatase. Same for others

13. Lines 165-199, the study design needs detailed explanation. Authors shoud provide additional information about the field trials.

  1. There is no information when and which plant growth stage, foliar spraying was performed. Explain each step in detail.

15. Authors should include images of plant morphology and cellular changes in tissues for applied treatments (if feasible).

16. The discussion frequently repeats findings already presented in the results without thoroughly explaining the underlying mechanisms.

17. The conclusion does not adequately summarize the key findings of their research results.

  1. Figure 4. Why there is no statistical lattering on the bars
  2. Figure 6, it is no clear which plant part is infected with late blight and which part is denoted as control. Author should highlight the targeted areas.

Author Response

1. The topic of research is not compelling. Authors should write proper title of this manuscript. 3. Authors should focus on the key findings of their study. 4. In abstract, there in no information how treatments were applied. A quantitative description of the percentage changes is missing. 6. The closing statement is very weak. The abstract doesn't provide a strong conclusion on the significance of the study. 7. In introduction, the first paragraph is rough and most of the provided information is not necessary. The lack of smooth transitions makes it difficult to follow the main argument 10. The objective stated is broad and doesn't clearly highlight the novelty or significance of the study. It does not tell the readers what the expectation is, or why this is important. Clearly articulate the hypothesis or research question to guide the reader on the study's focus. 17. The conclusion does not adequately summarize the key findings of their research results.
We have substantially revised the Abstract, Introduction, and Conclusion. We hope, the revised text more clearly supports the topic of the research.

2. Graphical abstract is very short and unclear. It is suggested to include plant image and highligh the key findings in graphical abstract.
We have comprehensively redesigned and improved all figures (Graphical Abstract, diagrams, charts) for better clarity and relevance.

5. The reader is left with a list of findings rather than a clear understanding of the overall mechanism by which silver nanoparticlessuppress disease severity in tomato plants. Highlight the significance of findings while maintaining scientific rigor. 8. Authors should elaborate on the siglaing or defense mechanisms of various nanomaterials in managing the pathogen infestation and then clearly establish the study's objective, demonstrating how this research advances beyond and differs from prior investigations 16. The discussion frequently repeats findings already presented in the results without thoroughly explaining the underlying mechanisms.
We have clarified and expanded mechanistic hypothesis, page 18: 565-586. We have expanded the literature review on AgNPs and oxidative stress; integrated a clearer, step-wise mechanistic hypothesis linking Ag⁺ release, ethylene receptor inhibition, hormonal pathway crosstalk (SA/JA/ET), and the resulting peroxidase/catalase dynamics. 

9. Please ensure that the full names of crops, NPs, and metals are used upon their first mention in the text. Subsequently, abbreviated or short names can be employed. 18. Figure 4. Why there is no statistical lattering on the bars
19. Figure 6, it is no clear which plant part is infected with late blight and which part is denoted as control. Author should highlight the targeted areas.
We have corrected the typographical errors, unified terminology, and improved overall formatting and figures.

11. Materials and methods. Keep the units same. There are few typo errors in this section. Kindly recheck in whole manuscript.
In materials andmethods, mention in detail which plant part was taken for the determination of assays such as peoxidase and acatase. Same for others 13. Lines 165-199, the study design needs detailed explanation. Authors shoud provide additional information about the field trials.
There is no information when and which plant growth stage, foliar spraying was performed. Explain each step in detail.
15. Authors should include images of plant morphology and cellular changes in tissues for applied treatments (if feasible).
We have provided clearer descriptions of field trial protocols, pages 4-6: 167-226.

Reviewer 4 Report

Comments and Suggestions for Authors

This study investigates the impact of polymer-stabilized silver nanoparticles (AgNPs) on the antioxidant enzyme activity in potato plants and their resistance against two pathogens, Phytophthora infestans and Alternaria solani. The research design is well-structured, integrating field trials with biochemical analyses. The data generally support the conclusions, and the work holds both innovation and practical relevance. The manuscript is written clearly and is logically organized, though certain details require further refinement.

1.Figures

The current layout of the figures appears somewhat disorganized. It is recommended that related panels be merged into composite figures to improve visual coherence and presentation.

Figure 9 (mechanistic diagram) is overly complex and visually cluttered. Consider simplifying the schema or presenting it in a stepwise manner for better readability.

Figures 9 and 10 present extensive mechanistic speculation regarding SA/JA/ET signaling, ROS dynamics, and related gene expression. However, the study does not include experimental measurements of these hormones, ROS, or gene expression levels, and the model is largely extrapolated from the literature. It is important to clearly label these parts as hypothetical or model-based inferences rather than direct experimental findings.

2.Terminology Consistency

The manuscript alternates between “peroxidase (POX)” and “peroxidase (POD).” Please unify the terminology throughout.

Similarly, terms such as “silver dispersions,” “silver-based treatments,” and “AgNPs dispersions” are used interchangeably. Consistent phrasing is recommended.

3.Environmental Fate and Accumulation of Ag

The study does not address the residue and accumulation of silver in plant tissues (leaves, tubers) or in soil. Given the agricultural application of AgNPs, a brief discussion on the potential environmental persistence, bioaccumulation, and effects on soil microbial communities should be included in the Discussion or Conclusion section. Even in the absence of original data, reference to relevant review articles on the environmental behavior of AgNPs would strengthen the manuscript’s contextual relevance.

Author Response

1.Figures
The current layout of the figures appears somewhat disorganized. It is recommended that related panels be merged into composite figures to improve visual coherence and presentation. Figure 9 (mechanistic diagram) is overly complex and visually cluttered. Consider simplifying the schema or presenting it in a stepwise manner for better readability. Figures 9 and 10 present extensive mechanistic speculation regarding SA/JA/ET signaling, ROS dynamics, and related gene expression. However, the study does not include experimental measurements of these hormones, ROS, or gene expression levels, and the model is largely extrapolated from the literature. It is important to clearly label these parts as hypothetical or model-based inferences rather than direct experimental findings.
We have completely redisigned Figures and labelled Figure 9 as based on literature data.

2.Terminology Consistency
The manuscript alternates between “peroxidase (POX)” and “peroxidase (POD).” Please unify the terminology throughout.
Similarly, terms such as “silver dispersions,” “silver-based treatments,” and “AgNPs dispersions” are used interchangeably. Consistent phrasing is recommended.
We have comprehensively improved all figures (Graphical Abstract, diagrams, charts) for better clarity and relevance; corrected the typographical errors, unified terminology, and improved overall formatting.

3.Environmental Fate and Accumulation of Ag
The study does not address the residue and accumulation of silver in plant tissues (leaves, tubers) or in soil. Given the agricultural application of AgNPs, a brief discussion on the potential environmental persistence, bioaccumulation, and effects on soil microbial communities should be included in the Discussion or Conclusion section. Even in the absence of original data, reference to relevant review articles on the environmental behavior of AgNPs would strengthen the manuscript’s contextual relevance.
We have added a brief discussion on AgNPs transformation in the environment, page 19-20: 603-628.
Additional discussion of AgNPs fate in the environment
In direct response to potential concerns regarding the environmental and food safety profile of our approach, we wish to highlight relevant prior research by our group. We have earlier conducted studies on wheat and apple trees treated with the same AgNPs formulations at comparable ultra-low doses (g/ha level). Using atomic absorption spectroscopy, we did not detect any significant accumulation of silver in the leaves, stems, fruits, or harvested grains at post-harvest. This evidence strongly supports the safety and environmental compatibility of the proposed application protocol and AgNPs, as the priming effect occurs without persistent residue.

Round 2

Reviewer 1 Report

Comments and Suggestions for Authors

The authors have improved the quality of paper and answer all the questions. I suggest the paper is ready for publication.

Author Response

We sincerely thank the reviewer for the careful evaluation of our manuscript and for the positive assessment of the revised version. We are grateful for the reviewer’s comments and suggestions, which have helped us improve the manuscript. We are pleased that the manuscript is now considered acceptable in its current form.

Reviewer 3 Report

Comments and Suggestions for Authors

All requested changes have been made. I recommend acceptance

Author Response

We sincerely thank the reviewer for the positive evaluation of our manuscript and for recommending its acceptance. We are pleased that the revised version has addressed all comments and concerns. We appreciate the reviewer’s time and constructive feedback, which helped to improve the quality and clarity of the manuscript.

Reviewer 4 Report

Comments and Suggestions for Authors

The revisions have addressed my concerns from the initial draft, and I find the manuscript acceptable in its current form.

Author Response

We sincerely thank the reviewer for the positive evaluation of our manuscript. We confirm that all concerns raised during the review process have been fully addressed, and the manuscript has been carefully revised accordingly. We are pleased that the revised version is now considered acceptable in its current form.